# hls4ml: An Open-Source Co-Design Workflow to Empower Scientific Low-Power Machine Learning Devices

## ABSTRACT

Accessible machine learning algorithms, software, and diagnostic tools for energy-efficient devices and systems are extremely valuable across a broad range of application domains. In scientific domains, real-time near-sensor processing can drastically improve experimental design and accelerate scientific discoveries. We have developed hls4ml, an open-source software-hardware co-design workflow to interpret and translate machine learning algorithms for implementation in FPGAs and ASICs specifically to support domain scientists. In this paper, we describe the essential features of the hls4ml workflow including network optimization techniques—such as pruning and quantization-aware training—which can be incorporated naturally into the device implementations. We expand on previous hls4ml work by extending capabilities and techniques towards low-power implementations and increased usability: new Python APIs, quantization-aware pruning, end-to-end FPGA workflows, long pipeline kernels for low power, and new device backends include an ASIC workflow. Taken together, these and continued efforts in hls4ml will arm a new generation of domain scientists with accessible, efficient, and powerful tools for machine-learning-accelerated discovery.

## 1 INTRODUCTION

Efficient implementations of machine learning (ML) algorithms in dedicated hardware devices at the "edge," or near-sensor, has numerous advantages. Edge processing and data compression can greatly reduce data rates and the energy required for data movement. Furthermore, real-time data processing and interpretation can greatly accelerate decision-making, hypothesis testing and even enable just-in-time interventions.

Staggering data rates and massive datasets are generated across a broad range of modern scientific applications in high energy physics, material science, and astrophysics. For example at the CERN Large Hadron Collider (LHC), experiments typically produce data at rates of Pb/s, and at the Fermilab accelerator complex, hundreds of thousands of devices monitor miles of beamlines that steer near speed-of-light particle beams. Low-latency ML is required for real-time decision making in these physics experiments with a range of requirements from tens of nanoseconds to sub-millisecond. In many ways, techniques for resource-constrained ML implementations are similar whether targeting low power or

Permission to make digital or hard copies of all or part of this work for personal or classroom use is granted without fee provided that copies are not made or distributed for profit or commercial advantage and that copies bear this notice and the full citation on the first page. Copyrights for components of this work owned by others than ACM must be honored. Abstracting with credit is permitted. To copy otherwise, or republish, to post on servers or to redistribute to lists, requires prior specific permission and/or a fee. Request permissions from permissions@acm.org.

*tinyML '21, March 22–26, 2021, Online*

© 2020 Association for Computing Machinery.

ACM ISBN 978-x-xxxx-xxxx-x/YY/MM. . . $15.00

https://doi.org/10.1145/nnnnnnn.nnnnnnn

**Unpublished working draft. Not for distribution.**

ultra low latency and high throughput. **In this paper, we discuss how tools developed for low-latency applications in science could be deployed for low-power applications.**

*Demand for accessible tools*. Low-power ML is in demand in scientific, industrial, and commercial computing [1]. Fitness bands, smartwatches, and other wearables that capture human health and behaviors from complex and multivariate continuous sensor data [2], wireless sensor networks deployed for tracking and understanding threatened animal populations using imaging and other sensors [3], and even large-scale wireless agriculture sensing [4] all necessitate powerful local computation on a budget. Despite the broad need for local, low-power ML and the growing number of edge devices and scientific applications, general-purpose off-the-shelf hardware has not kept pace with these computing demands. The challenge for domain scientists is that a broad range of expertise is required to arrive at full ML device implementations. This requires significant resources and a team of domain scientists, computer scientists, and engineers. Typically, domain scientists may be experts in domain ML, their experimental devices, or system engineering techniques, but very rarely in all requisite areas simultaneously.

To tackle this challenge, we need a framework which makes digital hardware implementations of ML algorithms more *accessible, interpretable, and (re)usable* to domain scientists. While ultimately hardware implementations require completely engineered solutions, allowing domain scientists to co-design algorithms based on their system and application constraints is extremely valuable in reducing engineering time and enabling faster design iterations. Optimizing this process will greatly reduce the *time to science*. Finally, to cater to both large experimental collaborations and smaller laboratory groups, the tools should be as open-source as possible.

*ML-hardware co-design tools*. Software like TensorFlow and PyTorch have democratized ML for scientists, lowering the time-to-science across domains. We aim to extend the ML workflow to efficient hardware implementations through the creation the hls4ml framework [5], an open-source co-design workflow. After a user trains their ML algorithms in common ML software frameworks, hls4ml translates them into digital implementations using high-level synthesis (HLS) tools for energy-efficient devices like FPGAs and ASICs. With the introduction of an open-source framework like hls4ml, "tinyML" techniques can be made accessible to nonexperts. The benefit of hls4ml is two-fold: it lets nonexperts create bespoke, cutting-edge ML accelerators for low-power and low-latency systems, and it lets nonexperts develop intuition about how their design choices affect system power consumption.

A number of recent results highlight the power of the hls4ml approach including support for quantization down to binary and ternary precision [6], pruning, tunable parallelization [5], boosted decision trees [7], quantization-aware training (QAT) [8], and graph

neural networks (NNs) [9]. The development of `hls4ml` is application driven. While originally its main target was low-latency applications, recent work has focused on opportunities for longer latency, low-power applications. Algorithm design with `hls4ml` involves the generation of custom firmware for a specified NN architecture. The customized design ensures an efficient use of resources that is essential to run in low-latency, resource-constrained systems, and is useful across a broader range of applications. **This paper reviews salient core features of `hls4ml` and extends previous work by presenting a number of recently added features that aid in targeting low-power systems**: initial results of quantization-aware pruning (QAP), full end-to-end workflows that embed `hls4ml` algorithms into Vitis Accel designs for Xilinx FPGAs, new matrix-vector kernels which are optimized for longer pipeline intervals, sparse operations, and low power, and support for multiple HLS compilers and devices, including ASICs. Our approach for low-power devices focuses on ML for field-programmable gate arrays (FPGAs) and application-specific integrated circuits (ASICs) as energy efficient hardware architectures [10, 11]. The `hls4ml` framework aims to bridge novel techniques in NN inference optimization and hardware implementation while making it accessible to domain scientists.

This paper is organized as follows. In Sec. 2, we discuss the `hls4ml` workflow and features for introspection, validation, and support for multiple device types. In Sec. 3, we discuss co-design techniques developed at ML training to develop optimal hardware implementations. In Sec. 4, we describe how those NNs get implemented in hardware and the available configurations to the user. Finally, we summarize and present an outlook in Sec. 5.

## Related Work

There are other open-source efforts have explored ML on edge devices, including FPGAs, mobile devices, and microcontrollers with integrated workflows from training to deployment. The FINN project [12] is a framework from Xilinx Research Labs to explore quantized deep NN inference on FPGAs, with emphasis on generating dataflow-style architectures customized for each network. It includes tools for training quantized NNs such as Brevitas [13], the FINN compiler, and the finn-hlslib Vivado HLS library of FPGA components for QNNs. Further, TensorFlow Lite [14] is a set of tools to help developers run TensorFlow [15] models on mobile, embedded, and internet of things (IoT) devices. It currently supports Android, iOS, and Linux devices (like Raspberry Pi), as well as microcontrollers (like Arduinos). It enables on-device ML inference with low latency and a small binary size.

## 2 HLS4ML WORKFLOW

The task of automatically translating a trained NN, specified by the model's architecture, weights, and biases, into HLS code is performed by the `hls4ml` package. A schematic of a typical workflow is illustrated in Fig. 1. The first part of the workflow illustrated in red depicts the usual steps required to design a NN for a specific task. This component, performed with tools like (Q)Keras and PyTorch, involves a training step and possible compression steps (more discussion below in Sec. 3) before converging on a final model. The blue section of the workflow is performed with `hls4ml`,

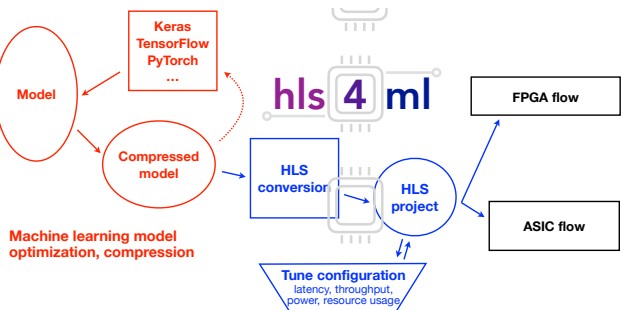

Figure 1: A typical workflow to translate an ML model into an FPGA or ASIC implementation using `hls4ml`. The red boxes (left) describe the model training and compression steps performed within conventional ML software frameworks. The `hls4ml` configuration and conversion steps are shown in the blue boxes (center). The black boxes (right) illustrate possible ways to export and integrate the HLS project into a larger hardware design.

which translates a model into an HLS project that can subsequently be synthesized and implemented on an FPGA or ASIC, as depicted by the black section.

At a high level, FPGA and ASIC algorithm design is different from programming a CPU in that independent operations may run fully in parallel or concurrently. Furthermore, independent operations may be pipelined such that the algorithm can accept new inputs while it is still operating on previous inputs. However, such operations consume dedicated resources onboard the FPGA or ASIC and cannot be dynamically remapped while running. The challenge in creating an optimal digital design is to balance available resources with achieving the power, latency, throughput goals of the target algorithm.

The `hls4ml` framework provides a number of configurable parameters which can help the user explore and customize the space of latency, throughput, power, and resource usage tradeoffs for their application. Because every application is different, the goal of the `hls4ml` package is to empower the user to perform this optimization through automated NN translation and design iteration. `hls4ml` leverages HLS to generate hardware modules from code written in high-level programming languages like C/C++ [16]. Each layer and activation type is implemented as a separate configurable module customized to perform that specific operation. During the `hls4ml` conversion, these modules are composed in the correct way to produce a full ML model. Large throughput and low latency can be achieved by pipelining data through the network. Furthermore, resource usage can be optimized because each layer is tailored during conversion to the specific model and, if desired, set of weights. This optimization extends to zero suppression, where the layer can be configured to skip multiplications by zero weights. Although it may lead to slightly less optimal performance than RTL-based design, HLS-based design has significant benefits: it raises the level of abstraction, reduces the iteration time, simplifies the validation phase, and enables greater exploration and evaluation of design alternatives.

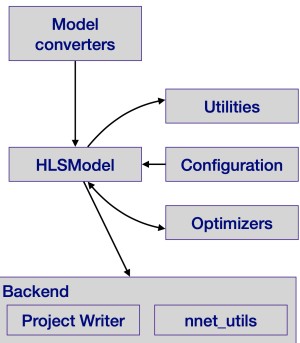

Figure 2: Internal structure of the `hls4ml` package. Model converters translate models from KERAS, PYTORCH, etc. into an intermediate HLSModel representation. This representation can be further configured and optimized. Different backend writers can be used to export the model into a given vendor-specific language, such as Vitis HLS, Quartus HLS, Catapult HLS, or others.

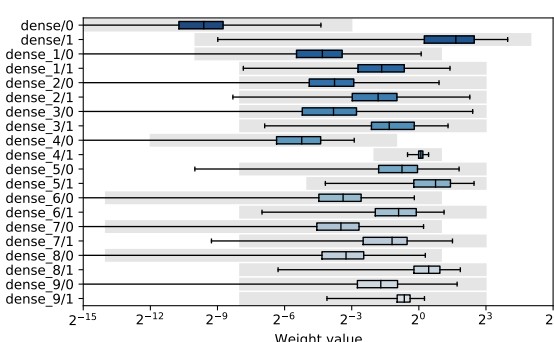

Figure 3: Numerical profiling plot from `hls4ml` for a fully-connected neural network. The distribution of the absolute value of the weights is shown on the x-axis. The items on the y-axis are the different weights (0) and biases (1) for the model layers.

## Package Architecture

To provide flexibility and ease-of-use, we implemented `hls4ml` as a PYTHON package that provides both a programming API and visualization capabilities. Figure 2 shows the internal structure of the `hls4ml` PYTHON package. The package first converts the user-specified model into a common internal representation of the network graph. Converters are provided for (Q)KERAS, TENSOR-FLOW, PYTORCH, and ONNX model formats. At the conversion step, the user-provided configuration is also attached to the model. For a NN trained with QKERAS quantization-aware training (QAT), the quantization settings of the model are propagated into the `hls4ml` internal representation.

A suite of *optimizers* then modify the network graph to target a more lightweight, faster inference. At this step, for example, batch normalization [17] layers are fused with the preceding dense or convolutional layer, or with the subsequent binary (or ternary) tanh activation layer. Where possible, these optimizations precompute quantities needed during inference involving constant model parameters, in order to reduce operations at runtime.

The `hls4ml` model object can be inspected, and the package provides a set of utilities to aid the configuration process. These include a visualization tool to display the NN graph decorated with the applied user configuration, and tools to numerically profile the model which can help guide the user settings, e.g. for bit precision.

An example of the numerical profiling output from `hls4ml` is shown in Figure 3 for a fully-connected NN. The distribution of the weight values is represented by a boxplot, showing the range covering the bulk of the distribution as well as the extremities. On top of this, the user-provided precision configuration is shown with the grey boxes. Generally, it is crucial that the largest absolute valued weights can be represented with the chosen type (the boxes overlap at the right of the plot). There is some flexibility to reduce the precision by truncating small valued weights, with minimal

impact on accuracy. This additional visualization tool can be used to quickly tune the configuration for more efficient inference.

One key feature of the programming API is the capability to execute the bit-accurate emulation of the generated HLS-synthesizable code in the PYTHON environment, for example as a Jupyter Notebook. In conventional HLS-design flows, developers craft C/C++ testbenches which they execute in the HLS-vendor simulation environment to verify algorithm performance. The `hls4ml` API enables a workflow that will be much more familiar to ML developers, where inference can be performed on tensor or array data in PYTHON code, providing the opportunity to complete detailed analysis. In addition to evaluating the `hls4ml` model output, users can access the detailed output of any hidden layer of the network, which can aid in debugging and performing hyperparameter optimization for quantized models. When the `hls4ml` model is written out, the backend maps the graph onto its library of optimized inference code. This inference code can run on the CPU executing the conversion, in order to check numerical correctness against the original NN. After that step, the user runs the vendor synthesis tools in order to produce an IP core, and evaluate latency, throughput, and resources. Presently, the most advanced backend is for Xilinx Vivado HLS, with codebases optimized for Intel Quartus HLS [18] and Mentor Catapult HLS [19] under active development.

## 3 NEURAL NETWORK TRAINING AND OPTIMIZATION

Reducing the precision of the calculations in the NN and removing unimportant calculations can drastically improve the efficiency of the NN implementation with little to no loss in performance. While applying these changes to a model post-training can be successful, to be maximally effective, we should consider these effects at the time of NN training.

We consider two benchmark tasks to demonstrate the versatility of model optimization in the `hls4ml` framework. The first is a high-energy particle jet classification task on a dataset [5, 20, 21] consisting of 16 features for simulated particle jets produced in

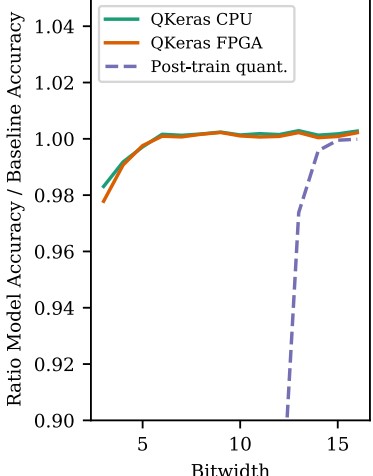

**Figure 4: Performance of quantization-aware training from Ref. [8] in terms of the relative accuracy as a function of bit width. The relative accuracy is evaluated with respect to the floating-point baseline model. The CPU-based emulation (solid green) of the FPGA-based QAT model (solid orange) is compared to the PTQ model (dashed purple).**

proton-proton collisions and originating from one of five classes of particles: W or Z bosons, light quarks, top quarks, or gluons. The baseline model is a simple fully-connected NN with three hidden layers, with 64, 32, and 32 nodes, respectively, activated by a rectified linear units (ReLUs) [22, 23]. The second benchmark is the MNIST handwritten digit classification task [24]. The baseline model we consider is a fully-connected NN with one hidden layer with 16 nodes and ReLU activation.

### 3.1 Quantization-Aware Training

Quantized [25–30] and even binarized [6, 28, 31–34] NNs have been studied as a way to compress NNs by reducing the number of bits required to represent each weight. FPGAs provide considerable freedom in the choice of data type and precision. Both choices should be considered carefully to prevent squandering FPGA resources and incurring additional latency. In `hls4ml`, we use fixed-point arithmetic, which uses less resources and has a lower latency than floating-point arithmetic. The inputs, weights, biases, sums, and outputs of each layer are all represented as fixed-point numbers. For each, the number of bits used to represent the integer and fractional part can be configured separately for the use case. The precision can be reduced significantly without causing a loss in performance [28]. We determine the number of bits to assign for the fractional part by scanning the performance values as a function of the bit precision.

One simple way to reduce a model's size is through post-training quantization (PTQ) where pre-trained model parameters are clipped or rounded to lower precision. However, this process is lossy and sacrifices model performance. To solve this, QAT has been proposed [35–37]. In these approaches, the reduced precision of the weights and biases are accounted for directly in the training of the

NN. In QKERAS, this is implemented using the straight-through estimator (STE) [31], where the forward pass of the training applies the quantization, while the backward pass assumes that quantization is the identity function, as the quantization function is not differentiable. It has been found that QAT is even more efficient than PTQ while retaining the same performance. In these studies, the same type of quantization is applied everywhere. More recently [38, 39], it has been suggested that some layers may accommodate extreme quantization better than other layers, suggesting that per-layer heterogeneous quantization is the optimal way to achieve high accuracy at low resource cost.

An example of the power of QAT is shown in Fig. 4 from Ref. [8] which uses QKERAS. For the particle physics task with a fully-connected NN, the accuracy of the reduced precision model is compared to the 32-bit floating-point implementation as the bit width is scanned. In the PTQ case, the accuracy begins to drop below 14 bits, while in the QAT case the accuracy is comparable to the 32-bit floating implementation down to 6 bits. More detailed discussion on layer-by-layer quantization is presented in Ref. [8]. In Section 4, we discuss the implementation of QAT in `hls4ml` and its effect in terms of on-chip resources.

### 3.2 Quantization-Aware Pruning

Network compression is a common technique to reduce the size, energy consumption, and overtraining of deep NNs [29]. Several approaches have been successfully deployed to compress networks [40–42]. Here we focus specifically on *parameter pruning*: the selective removal of weights based on a particular ranking [29, 43–47].

Prior studies have combined pruning and quantization trivially: by pruning 32-bit floating-point models and applying post-training quantization. One such approach, whose results are shown in Sec. 4.2, consists of iterative parameter pruning and retraining of a 32-bit floating-point model [5, 29, 48] with $L_1$ regularization, where the loss function is augmented with an additional penalty term $\lambda \|w\|_1$, where $w$ is a vector of all of the model weights and $\lambda$ is a tunable hyperparameter. $L_1$ regularization produces sparse models, provides built-in feature selection [49], and is readily available in many ML workflows. After training the model with $L_1$ regularization with a small $\lambda$ (e.g. $10^{-4}$), the weights are sorted based on their absolute value relative to the maximum absolute value of the weights in that particular layer. Weights falling below a certain percentile are removed. The model can then be trained again with $L_1$ regularization while masking the previously pruned weights. This processed can be iterated several times until reaching the desired level of compression.

While the above approach is effective, we describe here an alternative approach based on the lottery ticket (LT) hypothesis [45] where the remaining weights after each pruning step are initialized back to their original values ("weight rewinding"). We refer to this method as LT pruning. We also propose a new hybrid method for constructing efficient NNs, quantization-aware pruning (QAP), which combines a pruning procedure with training that accounts for quantized weights. As a first demonstration, we use Brevitas [13] to perform QAT and iteratively prune a fraction of the weights following the LT method of weight rewinding.

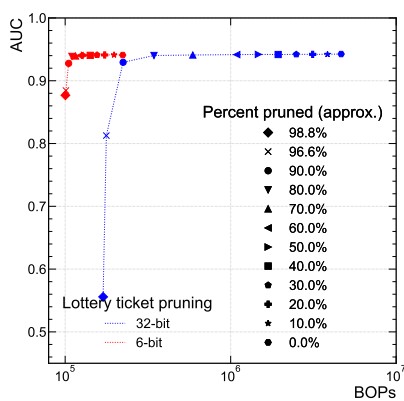

**Figure 5: Performance of quantization-aware pruning using the lottery ticket pruning scheme as a function of hardware computational complexity. After QAP, the 6-bit, 80% pruned model achieves a factor of 50 reduction in BOPs compared to the 32-bit, unpruned model with no loss in performance.**

This is done for the jet classification task presented in the previous section. At each training iteration, roughly 10% of the original network is pruned. The results of pruning with this method are shown in Fig. 5 for a 6-bit fixed-point version of the network compared to the 32-bit floating-point model. The performance in terms of the area under the curve (AUC) is shown as a function of bit operations (BOPs) [50], defined per-layer as

$$\text{BOPs} = mn((1 - f_p)b_a b_w + b_a + b_w + \log_2(n)) \quad (1)$$

where $n$ ($m$) is the number of inputs (outputs), $b_w$ ($b_a$) is the bit width of the weights (activations), and $f_p$ is the fraction of pruned layer weights. BOPs are a measure of the hardware computational complexity of a quantized NN after pruning. While in Sec. 3.1 we found that a 6-bit implementation of this network sacrificed no performance, we find here that pruning the 6-bit network by 80% using QAP still maintains the same performance as the 32-bit version.

## 4 DIGITAL IMPLEMENTATION ELEMENTS

Following the training-time optimizations described in the previous section, we describe important elements for deploying those optimized NNs in efficient digital implementations.

### 4.1 Quantization with a QKERAS Front-End

Reducing precision saves resources used for signal routing as well as resources and latency used for mathematical operations. As an example, the limiting resource for many FPGA applications is the number of DSPs, which are used primarily for multiplications. The number of DSPs used per multiplier depends on the precision of the numbers being multiplied and can change abruptly. For example, one Xilinx DSP48E1 block [51] can multiply a 25-bit number with an 18-bit number, but two are required to multiply a 25-bit number with a 19-bit number. Similarly, the latency of multipliers increases with precision, though they can remain pipelined.

| Model bit width | 14 | 6 |
|---|---|---|
| Accuracy [%] | 74.4 | 74.8 |
| Latency [ns] | 45 | 55 |
| DSP [%] | 56 (1826) | 1.8 (124) |
| LUT [%] | 5.2 (48321) | 3.4 (39782) |
| FF [%] | 0.8 (20132) | 0.3 (8128) |

**Table 1: Model accuracy, latency and resource utilization for 14-bit and 6-bit models. Resources are listed as a percentage of available resources, absolute numbers quoted in parenthesis, for a Xilinx Virtex UltraScale+ VU9P FPGA with a clock frequency of 200 MHz**

To allow for automated translation of a QKERAS model to register-transfer level (RTL), hls4ml has been extended to interpret and optimize quantized QKERAS layer types. When converting a QKERAS model into an HLS project, the model quantization configuration is passed to hls4ml and enforced in the FPGA firmware. This ensures that the use of specific, arbitrary precision in the QKERAS model is maintained during inference. For example, when using a quantizer with a given rescaling parameter $\alpha$, hls4ml inserts an operation to rescale the layer output. For binary and ternary weights and activations, the same strategies as in Ref. [6] are used. With binary layers, the arithmetical value of "-1" is encoded as "0," allowing the product to be expressed as an XNOR operation.

As an example of the integration of QKERAS and hls4ml, we now present an FPGA implementation of the model presented in Sec. 3.1. The FPGA implementation results are presented in Table 1 for the 14-bit PTQ and 6-bit QAT models. The effect of QAT is that the FPGA resources are drastically reduced, especially in the case of DSPs. In Ref. [8], a more detailed exploration of model implementations is presented, including per-layer optimizations.

### 4.2 Parallelization and Sparsity

The core component of dense and convolutional NN layer implementations in hls4ml is a matrix-vector multiplication kernel. In addition to the precision at which these kernels are executed there are further configurations that can be used to tune the digital design for a specific task. We explore two of them here: parallelization and sparsity.

*Parallelization.* A matrix-vector multiplication kernel can be characterized as a number of multiplication operations based on the dimensions of the matrix. The trade off between latency, throughput and FPGA resource usage is determined by the parallelization of the inference calculation and how many multiplications are performed in parallel. In hls4ml, this is configured with a "reuse factor" that sets the number of times a multiplier is used in the computation of a layer's output values. With a reuse factor of one, the computation is fully parallel, i.e. each multiplier is used once. With a reuse factor of $R$, $1/R$ of the computation is done at a time with a factor of $1/R$ fewer multipliers. To make routing more convenient, often there are preferred values of $R$ depending on the dimensions of the matrix itself.

The matrix-vector multiplication kernel cannot accept new inputs until all of the previous multiplications have been performed, a period of time known as the initiation interval (II). For larger reuse factors, the matrix-vector multiplication kernel will have a longer

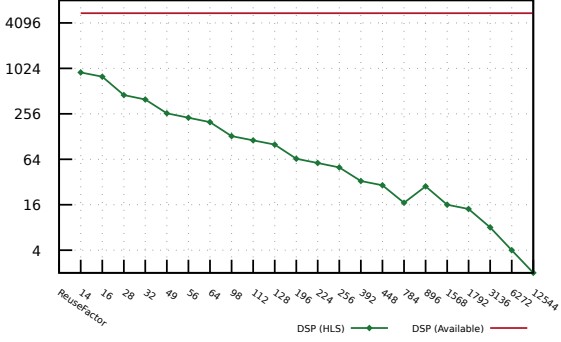

**Figure 6: DSP usage for the MNIST neural network implementation where the reuse factor $R$ is scanned. As $R$ is increased, the DSP usage decreases while the latency (not shown) increases accordingly.**

latency and II, but will use less on-chip resources. In `hls4ml`, we implement each layer calculation independently and sequentially. The calculation of one layer cannot be initiated until the calculation of the previous layer has completed. Therefore, the total latency is equal to the sum of latencies of each layer plus the latency required to connect the layers. The number of inferences per unit time (throughput) is inversely proportional to the reuse factor.

The configurability of the reuse factor allows users of `hls4ml` to tune their hardware implementation for their system requirements. In Fig. 6, we show the FPGA resources for a dense, fully-connected NN which is used for the MNIST handwritten digit classification task [24]. The total number of multiplications needed for this network is $(784)(16) + (16)(10) = 12,704$. The network is implemented in an FPGA with various reuse factors from 14 to $(784)(16) = 12,544$. In these implementations, the reduction in DSPs can be seen as $R$ is increased. Not shown in the figure is the complementary behavior where the latency and II of the NN increase commensurately. For example, the II of the NN increases from 14 to 12,544 clock cycles as $R$ increases. Thus, for a clock frequency of 100 MHz, the II of the network would increase from 140 ns to 0.125 ms.

*Sparse operations.* In Sec. 3.2, pruning is presented to create more efficient NN implementations by reducing the number of multiplication operations required to evaluate the network. By creating a network implementation where the matrix-vector kernel has a large fraction of zero-weights, the computation resources can be greatly reduced. In `hls4ml`, this can be built into the NN translation through a dedicated sparse matrix-vector multiplication kernel. There are two complementary implementations of this kernel in `hls4ml` depending on the size of the matrix and the latency of the operation required.

In the first implementation, HLS preprocessor directives are used to limit the number of multipliers available to the kernel based on the number of nonzero weights, and HLS is left to do the optimization. This is only feasible for smaller network layers. In the second implementation, the nonzero weights are compressed using a coordinate list (COO) representation where indices are packed into the weights themselves. The `hls4ml` user can specify a boolean

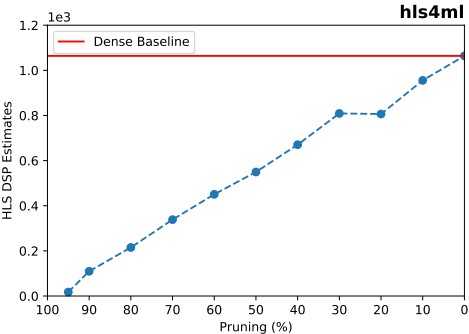

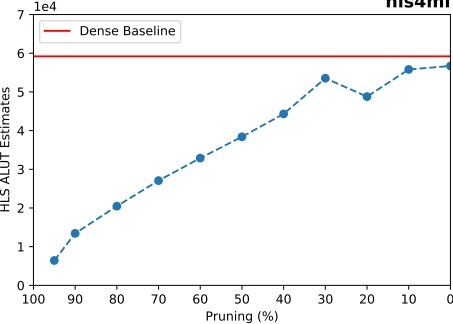

**Figure 7: DSP (upper) and LUT (lower) usage of the jet substructure classification network as a function of the percentage of the network pruned.**

`compression` parameter per layer, which activates this kernel. As an example, the COO implementation is used for the jet classifier NN described in previous sections. The architecture is pruned using an iterative pruning approach as described in Sec. 3.2 where the network is pruned in increments of 10% of the original number of network weights. Figure 7 illustrates the DSP and LUT usage of those NNs as a function of the pruned weight percentage. The figure shows the expected scaling of the pruned implementation where the resources decrease as a larger percentage of the network is pruned.

## 4.3 Device-Specific Workflows

*4.3.1 Xilinx FPGA workflow with Vitis.* There are multiple ways to execute an `hls4ml` project on a given FPGA. The RTL code created by Vivado HLS is fully functional and can be placed in a Vivado block design. While this allows the user to customize the implementation to meet specific design goals or to integrate the project into an existing firmware design, it can also present a barrier for less experienced developers. Vitis Accel is a Xilinx tool that aims to assist users in accelerating FPGA kernels. A Vitis Accel design consists of two components: the FPGA kernel and the host code typically run on a CPU. While the tool supports multiple kernel description languages, we have focused on HLS-based kernels. Vitis Accel imposes various constraints on the input and output of the kernel that requires us to introduce a wrapper around the default `hls4ml`

project. The host code is then able to manage the transfer data between the host CPU and the FPGA, either through DMA transfers or AXI streams. The choice of data transfer protocol is critical to the performance of the design. Typically, a small number of large data transfers is preferable to a large number of small data transfers. With SoC devices there is significant flexibility in customizing the data transfers due to the many different memory types available and their physical locations on the chip. Vitis Accel can be used to integrate hls4ml kernels. For smaller networks run with very large batch sizes, Vitis Accel and hls4ml are capable of producing highly performant accelerated co-processing workflows [52].

*4.3.2 ASIC workflow.* Domain scientists may choose an ASIC rather than an FPGA implementation when they aim at sacrificing reprogrammability for greater efficiency. However, designing for ASICs is significantly more complicated and time-consuming than for FPGAs. In the ASIC design workflow, verification and power analysis play a bigger role at the various levels of abstractions.

Figure 8 shows the ASIC workflow integrated with hls4ml. The ML training phase provides us with both the model and the stimuli for the subsequent verification steps. hls4ml compiles the trained model in a synthesizable C++ specification and a set of directives for Mentor Catapult HLS to target ASIC design [19]. Thanks to our state-of-the-art implementation of the C++ ML specification and optimized synthesis directives, the HLS-generated RTL code is comparable in power, performance, and area (PPA) to handwritten RTL [53]. ASIC design projects are often impractical for domain scientists because of the hardware's intrinsic complexity and the inability of many small-sized research groups to navigate the lengthy RTL-based hardware design cycle to meet acceptable deployment time frames. In our ASIC workflow, we can spend more time (1) refining the ML model thanks to the quick PPA estimation from Catapult HLS and (2) verifying both the C++ and RTL implementations to identify bugs and improve performance. We check design rules on the C++ specification by performing static analysis (Mentor CDesignChecker); we run C simulation and code coverage (Mentor CCov); finally, we run C&RTL co-simulation for equivalence checking [54]. The synthesized RTL code is subsequently processed with a traditional digital implementation flow that integrates simulation steps to ensure optimal PPA.

As a recent demonstration of this workflow, a completed design of a low-latency autoencoder for particle physics data compression has been implemented for the TSMC 65 ns technology node [55, 56]. The algorithm, trained in QKERAS, compresses on-sensor data with convolutional and dense layers to be transmitted off the detector. In order to maintain reconfigurability of the algorithm in changing experimental conditions, the weights can be updated via an I2C interface. The design also features triple modular redundancy to maintain its radiation tolerance up to 200 MRad. The algorithm, which has roughly 4,400 parameters, has a latency of 25 ns, is 3.6 mm$^2$ in area, and is estimated to consume 2.38 nJ per inference.

## 5 SUMMARY AND OUTLOOK

In this paper, we present the current status of the open-source co-design hls4ml workflow, focusing on new features and techniques relevant for low-power performance. We detail, for the first time,

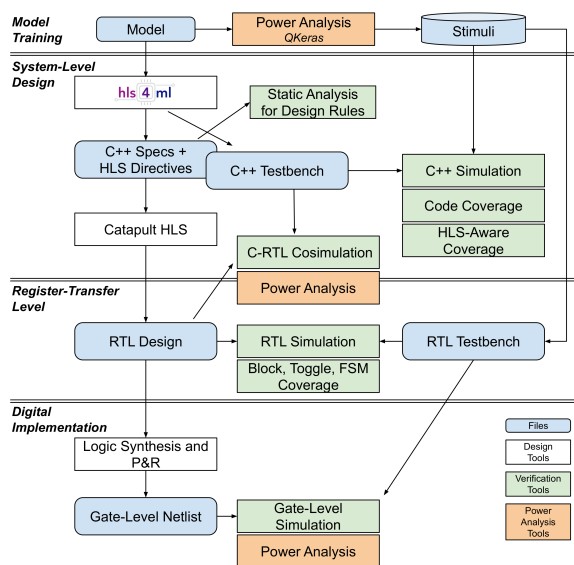

**Figure 8: Design and verification stack for the ASIC workflow.**

the structural features of hls4ml which allow for model introspection and validation in a PYTHON package and support for multiple device types. We also introduce quantization-aware pruning for neural networks building on previous work for quantization-aware training, providing additional significant resource savings. We also describe new hls4ml features for implementation for FPGAs and ASICs. These include configuration handles for quantization and pruning as well as for parallelization which can tune the algorithm from low latency to low power.

While the features of hls4ml presented in this paper provide already a set of powerful capabilities, the ultimate goal is to provide a complete end-to-end toolkit to empower domain scientists to design machine learning algorithms for low-power devices. This includes development based on dedicated domain-specific data sets, models, platforms, and existing implemented designs for a range of devices. Further maturation of introspection tools and workflows for design performance, validation, and close integration with power estimation into standard CAD tools will give neural network designers timely feedback about the power consumption of their design choices without needing to consult hardware experts. Effective design-space exploration from a hardware perspective allows domain scientists to optimize better the power-performance trade-offs of their systems. We hope that hls4ml will lead to broader adoption of machine learning techniques in the low-power regime in science, enhancing scientific research with tinyML.

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
