# OpenReview forum: "hls4ml: An Open-Source Co-Design Workflow to Empower Scientific Low-Power Machine Learning Devices"
_tinyml.org/tinyML/2021/Research_Symposium — tinyML 2021 Regular_

### Official Review · AnonReviewer2 · 2021-01-24

**Overall Merit Score:** 3

**Brief Summary:**

This paper explains the workings of hls4ml.  hls4ml is an open source tool used as a layer between standard (TensorFlow, Pytorch) ML tool flow and RTL HLS (high level synthesis).  This tool will allow users who do not have the requisite skill needed for a HW flow to target FPGAs and ASICs.

The paper has a discussion and some of the considerations for a target on the FPGA/ASIC that many without the HW background may not know.

hls4ml has some capabilities for quantization and pruning to optimize the network in terms of compute and memory before targeting to HW.

**Detailed Comments:**

in section of strengths and weaknesses.

**Paper Strengths:**



- Figure 3 shows a visualization available.  It is good to have such a visualization on a tool like this.
- There is a discussion about debugging and the capabilities available in this flow.  Support for bit-level accuracy is good.  A lot of people underestimate the results of targeting HW.  It is very difficult to debug without bit level accuracy.
- There are concrete examples of results.

**Paper Weaknesses:**



- It would be beneficial to see some metrics of a design targeted with hls4ml vs. a hand coded design.  Even a layer would be instructive to understand the "costs" of this automation in terms of clock speed, area/memory, throughput, etc.
- Figure 3 only has dense (FC) layers.  These are less typical for the entire network.  Was there a reason a convolutional or pointwise layer was not shown.  Also, it would be beneficial to see a branching network such as ResNet.
- Figure 5.  This figure shows the comparison o BOPs vs. AUC.  This is an important comparison on a processor like design.  For a target of an FPGA (or ASIC) a more appropriate comparison would be size (e.g. LUT, utilization, latency, frequency) vs. AUC would be more appropriate.  If the target had lower BOP but ended up with more idle multiplier in the design, it would lead to a larger and potentially slow design while also having a lower AUC.
- Table 1.  Would like to see a 16-but comparison.  For example in most DSPs, a 14-bit implementation would have been more cumbersome since the native type may be 16-bits.  Also, on most FPGAs, the fabric multiplier is not 14 bits. It is often 16 (two 8 bits) or 18 bits (two 9 bits).  Depending on routing, a 18 bit design may be no larger than a 14 bit design and yield better accuracy.
- I would like to see the network (even a cartoon) of the network that was being targeted outside of Figure 3.

**Poster (If Paper Is Rejected):**

1: Yes, ok for poster sesion to nurture work

**Reviewer Confidence:**

4: The reviewer is confident but not absolutely certain that the evaluation is correct

---

### Official Review · AnonReviewer1 · 2021-01-27

**Overall Merit Score:** 4

**Brief Summary:**

This paper presents HLS4ML ,an open-source tool (Python Library)  for High-Level Synthesis, whether it be ASIC or FPGA flow for tunable Latency vs Power options.

This tool leverages a few other tools under the hood, notably Qkeras for quantization/pruning (where it too notably leverages TensorFlow framework), and Catapault for logic synthesis.

The HLS4ML framework sets out to democratize experiments for the scientific community such that they can evaluate ML algorithms in combination with a real synthesizable hardware.
The paper goes on to illustrate its capabilities in handling the trained NN model in terms of fusing (Batch normalization layers), quantizing, pruning and then, during mapping to hardware, the user is also given a handle to choose mapping which achieves their desired tradeoff between latency, area and power.


**Detailed Comments:**

This paper illustrates the power of re-use in the community. As it sets out to do, it democratizes the ML process yet further, by bringing together :

-	Model training
-	Model Optimization
-	Model Mapping
-	Model Tuning in terms of not only precision but also hardware tradeoffs

As such, it shows a framework which can enable a lot of researchers to have at their fingertips the knowledge encapsulated from many domain specific experts (Quantization, pruning, High-Level-Synth, Graph optimization).


**Paper Strengths:**

HLS4ML is a very a good initiative which aims to pull together various resources and make them accessible for the community under one hood.
Just as tool sets such as Keras have shown, it is possible to abstract ML training frameworks such as TensorFlow, to a much higher level of abstraction, thus making ML technology more wieldy and accessible to users who prefer to operate at a higher level abstraction.
Having a tool which further leverages on existing frameworks (eg TensorFlow and Keras, Catapault HLS etc) is a powerful layer of abstraction which allows the user to even more easily define algorithms and explore their impact on actual final hardware implementation.


**Paper Weaknesses:**

It would be nice to see some actual hardware that the tool generated and get an idea of size/architecture.

**Poster (If Paper Is Rejected):**

1: Yes, ok for poster sesion to nurture work

**Reviewer Confidence:**

5: The reviewer is absolutely certain that the evaluation is correct and very familiar with the relevant literature

---

### Official Review · AnonReviewer4 · 2021-01-29

**Overall Merit Score:** 2

**Brief Summary:**

This paper argues for tools to generate FPGA NN accelerators to enable domain experts to accelerate their workloads.  The proposed tool hls4ml leverages commercial HLS tools and known model optimizations such as quantization and pruning.  A few point results are given on MNIST.

**Detailed Comments:**

I can see the motivation for this work, but in its current form, there's minimal contribution and can't really recommend this in its current form.  It's necessary to show a non-trivial example (>> MNIST) and compare the results against previously published designs to allow the reader to understand what the trade-off is between HLS and RTL design.  I also wasn't exactly clear on how the C code generation works - is this an overlay machine, or a completely flat implementation of the model?  It would have been nice to see some comparison with published accelerators - does hls4ml match hand implementations?  Finally, the related work section is very weak (just one paragraph), and missing a number of key references, some notably FixyNN (Whatmough et al, MLSys'19)



**Paper Strengths:**

Potentially a useful tool assuming that the user needs fast design time AND cares about PPA

**Paper Weaknesses:**

Low novelty, model optimizations are well known and no hardware novelty
Trivial and incomplete example (MNIST)
No comparison with previous work, e.g., FixyNN and others



**Poster (If Paper Is Rejected):**

1: Yes, ok for poster sesion to nurture work

**Reviewer Confidence:**

4: The reviewer is confident but not absolutely certain that the evaluation is correct

---

### Official Review · AnonReviewer3 · 2021-01-30

**Overall Merit Score:** 2

**Brief Summary:**

This paper goes through the workflow of hls4ml, which is a tool that enables a fast algorithm-to-hardware process. It incorporates several network optimization techniques and provides high-level APIs, which makes it powerful and user-friendly.

**Detailed Comments:**

hls4ml is a very solid tool that should prove it useful for many researchers. But this paper reads more like a technical report. Most of the applied techniques have been proposed in other literatures and exist in other tools. While it is certainly more convenient to have them all in one turn-key workflow, there lacks insight to objectively evaluate its technical advantages. For example, I’d certainly like to know if there are specific challenges that this workflow overcomes or if it further uncovers other co-optimization opportunities.

Also, there lacks objective evaluations on the performance and efficiency of the final implementations from the workflow with common benchmarks. For example, how do the PPA metrics of implementations from this flow compared to manually optimized designs (the paper only mentions that it is comparable with one design without qualitative evidence)? How does it optimize data movements, which is key to the efficiency of the hardware, in addition to compute and arithmetic in order to deal with increasingly complex network topologies? What are the specific strategies that it uses to tune for PPA, especially under the edge and near-sensor use cases where resources can be very limited?

From a tooling’s point of view, many existing frameworks and tools also start to incorporate the capabilities to do QAT, and there are also tools for QAP. What are the pros and cons of the proposed pipeline compared to the existing tools? How does it evaluate the efficiency of quantization schemes versus the efficiency of the result hardware?


**Paper Strengths:**

•	The tool provides a great utility that streamlines several optimization techniques to enable a fast acceleration experience

**Paper Weaknesses:**

- There are no fundamental technical improvements over state-of-the-art; it reads more like a technical report rather than a paper.
- There lack objective evaluations with common benchmarks on the quality of implementations from this flow.


**Poster (If Paper Is Rejected):**

1: Yes, ok for poster sesion to nurture work

**Reviewer Confidence:**

3: The reviewer is fairly confident that the evaluation is correct

---

### Decision · Program_Chairs · 2021-02-05

**Decision:**

Accept (Regular)

**Comment:**

Congratulations on your paper's acceptance!

Your paper has been accepted as a full-length regular paper.

Please read the reviews carefully and make sure the concerns are addressed in your final submission.

All accepted papers will be given a slot in the TinyML Summit schedule for an oral presentation on Friday, March 26, 2021.

Camera ready instructions will follow soon. All papers will be hosted on arXiv and published papers will have the following header stamp: “Published as a conference paper at TinyML Research Symposium 2021.” The paper will also be presented on the program website.